# Biological Embedding of Early-Life Adversity and a Scoping Review of the Evidence for Intergenerational Epigenetic Transmission of Stress and Trauma in Humans

**DOI:** 10.3390/genes14081639

**Published:** 2023-08-17

**Authors:** Aoshuang Zhou, Joanne Ryan

**Affiliations:** 1Division of Epidemiology, Jockey Club School of Public Health and Primary Care, Chinese University of Hong Kong, Hong Kong SAR, China; 2School of Public Health and Preventive Medicine, Monash University, Melbourne, VIC 3004, Australia

**Keywords:** intergenerational, transgenerational, epigenetics, trauma, stress, post-traumatic stress disorder

## Abstract

Severe or chronic stress and trauma can have a detrimental impact on health. Evidence suggests that early-life adversity can become biologically embedded and has the potential to influence health outcomes decades later. Epigenetics is one mechanism that has been implicated in these long-lasting effects. Observational studies in humans indicate that the effects of stress could even persist across generations, although whether or not epigenetic mechanisms are involved remains under debate. Here, we provide an overview of studies in animals and humans that demonstrate the effects of early-life stress on DNA methylation, one of the most widely studied epigenetic mechanisms, and summarize findings from animal models demonstrating the involvement of epigenetics in the transmission of stress across generations. We then describe the results of a scoping review to determine the extent to which the terms intergenerational or transgenerational have been used in human studies investigating the transmission of trauma and stress via epigenetic mechanisms. We end with a discussion of key areas for future research to advance understanding of the role of epigenetics in the legacy effects of stress and trauma.

## 1. Introduction

### 1.1. Stress as a Major Risk Factor for a Range of Non-Communicable Diseases

Psychological stress is a feeling of immense emotional, mental, or physical pressure, and strain, which exceeds an individual’s capacity to adapt and respond. High levels of stress can result from a perceived acute or chronic stressful situation/circumstance, a major traumatic event, whether it is anticipated or unpredictable, or other severe adverse events in which an individual struggles to cope. It is normally accompanied by cognitive and behavioral changes, and a cascade of physiological systems is activated to help the individual deal with the situation [1]. Two endocrine systems are particularly responsive to stress. Activation of the sympathetic–adrenal–medullary system causes the release of catecholamine, which helps regulate the cardiovascular, immune, and respiratory systems, and the hypothalamic–pituitary–adrenal (HPA) axis, which results in the release of glucocorticoids (e.g., cortisol in humans). Tight regulation of the HPA axis is central to the stress response system with a negative feedback loop, ensuring that the system returns to a homeostatic balance when the perceived stress is no longer present. Periods of chronic or severe stress, however, have been shown to result in the dysregulation of the stress response system, and this can influence a broad range of physiological processes, including metabolic functioning and the immune system [2]. While activation of these systems is beneficial under normal circumstances, excessive or prolonged stimulation is maladaptive, resulting in adverse biological effects on the body and brain. Indeed, it is now clear that stress is an important risk factor for a range of diseases [3,4].

Stress and trauma are also risk factors for a range of non-communicable diseases, including cardiovascular disease [5,6], diabetes [7], and cognitive impairment [8,9], and can contribute to cancer incidence and survival [10]. Solid evidence also shows that stress and trauma are important risk factors for a number of mental health conditions. The etiology of post-traumatic stress disorder (PTSD), the most severe and incapacitating anxiety disorder, is conditional on having been exposed to a traumatic event. Dysfunction of the HPA axis and blunted cortisol response has been found in adults with PTSD [11,12]. Stress is also thought to play a key role in the pathogenesis of depression. Depressive episodes are frequently triggered by stressful events, and stress can worsen the clinical course of depression [13,14]. Major depressive disorder has been associated with increased cortisol levels and impaired feedback sensitivity of the HPA axis [15]. HPA axis dysfunction may actually be a permanent trait of depression, with hyper-secretion of diurnal cortisol even after remission of depressive symptoms [16].

The negative impact of stress and trauma on physical and mental health, however, has been shown to vary across individuals. This is likely to depend on the severity of the exposure, the subsequent lifestyle and behavioral choices [17], and differing genetic vulnerability [18,19,20]. The timing of exposure is also likely to play an important role. This review will focus on psychological stress and the potential impacts of the activation of the stress response systems. It is important to note, however, that the impact of stress can be much broader, and have flow-on effects that impact an individual’s health and lifestyle. For example, socioeconomic disadvantage is a chronic type of stress and an important risk factor for infectious diseases. The impact of this ‘stress’ on future generations could thus occur through mechanisms and pathways unrelated to the stress response system. This topic is beyond the scope of the current review and will not be detailed here.

### 1.2. Early-Life Programming

A large body of data highlights the importance of the early life environment in shaping our health and risk for disease in later life. Almost 40 years ago, the late epidemiologist David Barker first observed a strong correlation between infant and adult mortality rates in certain geographical regions of the United Kingdom [21] and later a link between low birth weight and risk of disease in later life [22]. These observations contributed to a growing theory referred to as fetal programming or the Barker Hypothesis, which postulated that the fetus adapts to its in utero environment to maximize growth and development, but such adaptations can have long-term consequences postnatally [23]. This expanded to the Developmental Origins of Health and Disease (DOHaD) concept, which describes how during early life, the environment induces changes in development that can have long-term impacts [24]. Indeed, it is now well-established that the early-life environment plays a critical role in the ‘programming’ of health, influencing later risk of disease [25]. Adversity during early life could thus lay the foundations for increased susceptibility and sensitivity to stress occurring at a later stage, as well as more widespread effects on health.

Stress and/or trauma occurring during critical periods of development from fertilization through early childhood is thought to be particularly detrimental, as these are periods of substantial growth and development where biological systems are being established and the brain is forming. Any effects during this period are, therefore, more likely to become embedded, with long-term consequences [25]. Although adaptations to the external environment are performed in a way to optimize growth and ‘success’, they may not be ideal for the long term and may actually have more deleterious effects. Therefore, adversity in this early phase can shape the development of the stress regulation system and thus could lead to long-term alterations in stress response throughout life.

Indeed, findings from numerous studies suggest that early-life trauma can result in dysregulation of the HPA axis, which in turn impacts numerous biological systems, including immune and neurotransmitter systems [26]. Abnormal cortisol secretion and blunted cortisol response have been reported following early-life stress [27]. Mounting evidence has also shown that early-life stress could increase the risk of a range of health problems (e.g., autoimmune disease, cardiovascular disease, depression, diabetes, hypertension), which can persist into adulthood [28,29,30].

Considerable recent advances in our understanding of the genome and how it is regulated provide solid support for the role of epigenetic mechanisms in helping explain how stress and trauma can become biologically embedded, and exposures early in life can have long-lasting health consequences [31].

### 1.3. Involvement of Epigenetic Mechanisms in the Biological Embedding of Stress

Epigenetic mechanisms are a suite of environmentally sensitive and potentially reversible modifications to the genome (‘above’ the level of the DNA), which can regulate gene activity without altering the genetic code/DNA sequence, leading to long-term changes [32]. These are essential for normal cell development, enable tissue differentiation in utero, and are highly dynamic early in development, responding and adapting to environmental stimuli but also influencing the underlying genetic code. This plasticity of the epigenome enables optimal adaptation and more favorable responses to certain environmental conditions through modifications of gene activity. The best-understood and most stable epigenetic mechanism is DNA methylation, which commonly involves the addition of a methyl group at CpG dinucleotides in the regulatory promoter regions of genes, thus influencing how a gene is expressed. Epigenetic modifications can remain relatively stable and thus have the capacity for long-term changes in biological processes.

Increasing evidence also suggests that the epigenome is likely a causal factor for, as well as a marker of, disease risk. Epigenetic disruption has been clearly implicated in cancer, aging, and a range of complex diseases, including psychiatric disorders, such as depression [33] and PTSD [34,35]. Epigenetic mechanisms are now widely accepted as playing an important role in the early-life programming of disease risk [36]. Exposure to early-life stress and trauma is likely to impact the epigenome, and epigenetic alterations may help explain the long-term effects of trauma on later health.

#### 1.3.1. Animal Studies of Early-Life Stress

Animal models provide a useful tool for examining epigenetic effects in response to trauma by providing an experimental design that is not possible in human studies. Indeed, the vast majority of research in this area has stemmed from a seminal study in rats by Meaney and colleagues published almost two decades ago, demonstrating that early-life stress was associated with glucocorticoid receptor (*Nr3c1*) methylation [37]. Rat pups exposed to low levels of maternal care (i.e., reduced maternal licking and grooming) had disrupted HPA axis signaling and differential *Nr3c1* DNA methylation in the hippocampus, a brain region particularly implicated in memory and emotional reactivity [37]. Furthermore, these effects persisted into adulthood but could be reversed through cross-fostering the pups to a female who displayed high levels of maternal care or after administration of a chemical compound to reverse the epigenetic marks. This indicated that the findings were not simply explained by genetic factors. This important work opened up a whole new area of research into the role of epigenetics in the long-term effects of early-life adversity.

Numerous subsequent studies using rodent models have shown that early-life trauma (e.g., chronic and unpredictable maternal behavior, separation, fear conditioning), can influence DNA methylation of various candidate genes in brain regions important in stress signaling. For example, prenatally stressed rats were shown to have increased brain-derived neurotrophic factor (*Bdnf*) promoter methylation and correspondingly decreased expression in the amygdala and hippocampus [38]. This was observed in early life (at weaning) and persisted into adulthood. Other candidate genes shown to be differentially methylated include reelin (*Reln*) [39], which is involved in neurogenesis and synaptic plasticity, corticotropin-releasing hormone receptor (*Crh*) [40], which binds corticotropin-releasing hormone and plays an important role in stress signaling, and arginine vasopressin gene (*Avp*) [41]. Increased DNA methylation levels of *Gdnf* (Glial cell-derived neurotrophic factor) and *Crh* in rodent models of chronic stress have been observed, although the effects appear to be both gender and brain region-dependent [42,43].

#### 1.3.2. Post-Mortem Human Studies

One of the first studies to show clear evidence of an association between trauma and epigenetic disruption in humans was a post-mortem study of the brain tissues of 36 adult suicide victims [44]. McGowan and colleagues found that a history of severe childhood abuse (ascertained with proxy-based interviews) was associated with increased *NR3C1* methylation in the hippocampus. A subsequent hypothesis-free epigenome-wide association study (EWAS), interrogating thousands of CpG sites across the epigenome, identified 362 genetic loci that were differentially methylated in the hippocampal brain tissues of 25 men with a history of severe early-life trauma compared to 16 controls [45]. The most significant finding concerned the Alsin gene (*ALS2*), which is involved in regulating small GTPase activity [45].

When investigating the possible involvement of epigenetic processes in stress and PTSD, human post-mortem studies of brain tissue are particularly relevant. Unlike genetic marks, which are the same across all tissues in the body, epigenetic patterns are tissue- and cell-specific, and the brain would thus be the most appropriate tissue for disorders involving central processes. However post-mortem studies are not without their own limitations, which include confounding related to the cause of death (including possible comorbid psychiatric disorders, like depression), the timing and condition under which the sample is obtained, the lack of detailed patient history, including information on the trauma itself and diagnosis of other psychiatric conditions, and the fact that findings from such studies will only ever be correlational (i.e., lacking a prospective design to help ascertain causality). Conversely, the usefulness of peripheral epigenetic markers in complex neurobiological phenotypes and brain disorders is now well-recognized [33]. This may be particularly relevant for trauma, which is known to impact a range of biological systems involving humoral mechanisms, including stress reactivity and the HPA axis, or associated inflammatory response. Furthermore, there may be a correlation between brain and blood methylation marks [46].

#### 1.3.3. Peripheral Biomarkers of Early-Life Stress

Numerous studies have now investigated the association between early-life trauma and peripheral epigenetic marks, demonstrating an association with methylation of candidate genes (e.g., *BDNF*), genes involved in synaptic plasticity, neurotransmission, and immune function [20], as well as novel genes [47,48]. For example, it was shown that adults subjected to childhood abuse had differential methylation in 997 genes, many of which were involved in developmental cell signaling pathways [49]. A genome-wide study of men found that parental physical aggression in childhood and adolescence was associated with differential methylation in genes previously implicated in aggressive behavior [50]. One notable study measured the *FKBP5* gene in blood, which encodes the FK506 binding protein 5, acts as a chaperone to the glucocorticoid receptor, and regulates glucocorticoid sensitivity. They found differential DNA methylation in response to childhood trauma, and this increased the risk of stress-related psychiatric disorders [51].

Much attention has again focused on *NR3C1* promoter DNA methylation [52,53]. Research has shown that adult’s retrospective reporting of parental loss, lack of parental care, or maltreatment in childhood was associated with increased *NR3C1* methylation, and this, in turn, was linked to an attenuated cortisol stress response [54]. However, the strongest evidence comes from studies of in uerto exposure to stress during development. There is now consistent evidence for a link between pregnancy stress (including perceived stress, anxiety, depression, and natural disasters) and increased DNA methylation of *NR3C1* in newborns [55,56], even though the studies have been quite heterogeneous in terms of the type of exposure and tissue investigated (i.e., cord blood, peripheral blood, placenta, buccal swabs). For example, increased methylation was found in newborns exposed to prenatal stress [57] and the cord blood of mothers with third trimester depression/anxiety [58]. The findings concerning *NR3C1* are of particular relevance given that they align with the animal studies and replicate the earlier observations with brain tissue. This suggests the possibility that at least for some genes, peripheral epigenetic patterns may be a good reflection of changes occurring in specific brain regions.

Overall, there is accumulating evidence that early-life stress, including stress in utero, can influence the offspring’s epigenetic patterns. Excitingly, there is also some preliminary evidence that exposure to stress preconception could also be transmitted across generations [59]. Epigenetics also provides a plausible mechanism to account for the findings that the effects of stress and trauma can be transmitted across generations.

### 1.4. Transmission of Stress across Generations

#### 1.4.1. Intergenerational and Transgenerational Inheritance

Increasing evidence suggests the potential that stress and trauma experienced by one generation can be passed down and influence the health of the subsequent generation/s. It is well-established in psychiatry that the offspring of parents with psychiatric disorders are themselves at an increased risk of behavioral and psychiatric problems [60,61]. For example, parents with PTSD have been shown to have offspring with more secondary traumatized symptoms compared to offspring of parents without PTSD [62]. There is also some preliminary evidence that exposures occurring even in earlier generations (e.g., grandparents) can also influence an offspring’s risk [63]. For example, a 30-year study of seventy-seven families over three generations found that biological offspring with two previous generations affected with major depressive disorder had the highest risk of having major depression themselves [64]. Likewise, grandchildren of Holocaust survivors have been shown to have a higher incidence of PTSD, depression, and anxiety [65]. In animal studies, there has also been some evidence that paternal postnatal trauma can have behavioral effects on or up to, the fourth generation [66].

To what extent these associations are driven by shared genetics, the shared environmental and behavioral factors, which can persist and run in families, or a combination of these, is yet to be determined. Furthermore, there is some exciting new research over the last decade suggesting the potential that epigenetic mechanisms could be involved. Intergenerational and transgenerational inheritance are distinct terms but are sometimes used interchangeably or in different ways when talking about this inheritance. To avoid ambiguity, the definitions we have used here [67,68] are as follows:

Intergenerational inheritance: transmission from one generation to the next through direct exposure (G0 to G1). In the most simplistic form, it can involve genes that are passed on from parents to offspring in classical Mendelian genetics. It could also include epigenetic changes resulting directly from, for example, maternal exposures during pregnancy, which directly impact the developing fetus. However, it could also include the transmission of ‘exposures’ occurring prior to conception in G0, which can then influence G1.

Transgenerational inheritance: this refers to the non-genetic transmission of traits from one generation to the next, which are not through direct exposure. It suggests that certain experiences or exposures acquired by an individual during their lifetime can be passed on to future generations. The word transgenerational is defined as acting across multiple generations and typically considers G0 to G2 transmission. It can include the inheritance of traits through epigenetic modifications.

This is a field of research that has developed considerably over the last decade. For example, in this period there have been almost 10,000 articles that mention ‘intergenerational’ in the last 10 years on PubMed, and 4500 transgenerational. Of the latter, 94% of the articles also include the word ‘trauma’, indicating that ‘transgenerational’ is most commonly used in this context. We will first review the evidence from animal models of the inter- or transgenerational epigenetic transmission of stress and then conduct a scoping review to assess how common these terms have been used in human studies in this area.

#### 1.4.2. Findings from Animal Studies

The vast majority of findings in this area have come from animal studies of stress, enabling the experimental control of exposures across multiple generations. They provide some compelling evidence for the transgenerational transmission and inheritance of environmentally acquired risk factors across generations via the germline [69], with epigenetic mechanisms strongly implicated [70]. A detailed review of this comprehensive body of work is provided elsewhere (see, for example, [71,72]).

Early-life maltreatment of infant rats produced persistent changes in *Bdnf* DNA methylation that were passed to the next generation by maltreated mothers [73]. Furthermore, the findings suggested germ-line inheritance because cross-fostering could not completely erase these DNA methylation changes. Many studies of predator stress have found that methylation of *Nr3c1* genes and *Fkbp5* appear to play a role in the transmission of these effects to future generations [74]. These animal studies have provided unique insights into germ cell epigenetic changes in response to stress [75], most commonly involving male rodents. Extracellular vesicles exposed to stress had effects on the sperm and were observed in the next generation [76]. Likewise, paternal stress prior to conception can influence offspring behavior and DNA methylation patterns [77], and a mechanistic role for sperm microRNAs in the transgenerational transmission of stress reactivity to the offspring has also been demonstrated [70].

Exposure to stress or direct administration of the stress hormone corticosterone prior to conception results in numerous cellular and molecular modifications and alters the behavior of offspring [78]. For example, exposure to high levels of glucocorticoids can alter stress signaling systems and affect memory, and thus has implications for treating fear-related disorders [79]. In utero exposure to synthetic glucocorticoids in guinea pigs was shown to influence the cortisol response to stress in the offspring, and they had increased stress-related behaviors [80]. However, this effect was also observed in the next generation, with evidence of long-term programming. Effects on gene transcription were observed in both the medial prefrontal cortex and the hypothalamic paraventricular nucleus. A similar study investigated corticosterone administration in adult male mice, and male pups (F1) displayed a hyper-anxiety phenotype [81]. This also resulted in subsequent progeny (F2) with reduced anxiety but an increase in depressive behaviors [77,78,79,81]. Furthermore, these offspring phenotypes (F1 and F2) were associated with miRNA expression in the paternal sperm, which was induced in response to corticosterone exposure.

The results of studies using mouse models have also demonstrated the potential for environmental enrichment to prevent the increased risk of psychiatric disorders resulting from transgenerational exposure to trauma. Early postnatal exposure of male mice to traumatic stress resulted in offspring that were more capable of responding to adversity in adulthood, and these effects appeared to be epigenetically regulated [82]. The offspring had decreased DNA methylation and increased glucocorticoid expression in the hippocampus. Interestingly, if the father’s exposure occurred when they reached adulthood rather than in early life, no effect on offspring behavior was observed.

#### 1.4.3. Evidence in Humans—Need for a Scoping Review

Despite these exciting emerging findings from animal studies, it remains unclear to what extent these findings could be applied to humans. One of the first studies to demonstrate the involvement of epigenetic alterations in the intergenerational transmission of environmental exposure in humans was undertaken on a small cohort of Holocaust survivors (n = 32, mean age 78) and their adult offspring (n = 22, mean age 46), with eight matched unexposed parent–offspring dyads [83]. Holocaust exposure was associated with blood DNA methylation of the *FKBP5* gene at the same transcriptionally active site in both generations, although interestingly in opposing directions. Exposed parents had an average of 10% higher methylation compared to unexposed, while exposed offspring had 7.7% lower methylation. *FKBP5* regulates glucocorticoid receptor signaling, a central component of the HPA axis stress response. In their analysis, Yehuda and colleagues also controlled for current PTSD (51.6% of the Holocaust-exposed parents), childhood trauma in the offspring, or genetic variation in the gene. Potential epigenetic effects in maternal or paternal gametes could not, however, be investigated.

Here, we undertake a scoping review to determine the extent to which the terms intergenerational or transgenerational have been used in human studies, investigating the transmission of trauma and stress via epigenetic mechanisms.

## 2. Materials and Methods

### Search Strategy and Inclusion Criteria

A scoping review was conducted to determine the extent to which intergenerational and/or transgenerational epigenetic transmission of trauma or stress have been described in human studies. Scoping reviews are useful tools to scope a body of literature and clarify the extent to which concepts are being used through rigorous evidence synthesis. Although they do not require registration of a review protocol, they still require explicit information related to how articles were identified and selected for inclusion [84].

The bibliographic databases MEDLINE and EMBASE were searched using Ovid software on 29 June 2023, capturing all articles since database inception (1946 and 1947, respectively). The following combination of search terms was used: [epigenetic* (or) methylat* (or) histone* (or) *RNA*] (and) [transgeneration* (or) intergeneration*] (and) [trauma* (or) stress* (or) disaster (or) assault* (or) depress* (or) anxiety* (or) PTSD].

Eligible articles were those where research involved humans and were published in English. Review articles were excluded, as were case reports and case series, but all other primary research studies were eligible. Participants included in eligible studies were of any age or sex. Given the nature of the research question, it was anticipated that most studies would include at least two generations from one family, and often infants or young children were included.

Main exposure/intervention: any form of trauma or psychological stress. This definition was broad and could include mental stress, depression, anxiety, PTSD, a major natural disaster, physical, verbal, or sexual assault, a car accident, war trauma, or even biologically recognized criteria indicating high stress, such as raised cortisol levels. The mental health conditions did not need to meet formal psychiatric assessment as long as a valid instrument was used to assess recognized symptoms of the disorders. Self-report measures were also eligible for inclusion. The comparator in each study would be individuals who were not exposed to the trauma or stress under investigation.

Main outcome: epigenetic marks or mechanisms, which include:

DNA methylation, which commonly involves the enzymatic addition of a methyl group to the carbon 5 position of cytosine in a cytosine guanine (CpG) dinucleotide in humans. The majority of CpG dinucleotides are methylated, and these are scattered throughout the genome at a low density. However, genomic regions with a high density of CpG dinucleotides, termed CpG islands, are often found in the 5′ regulatory promoter regions of genes. DNA methylation in gene promoter regions can block the binding of transcription factors to reduce gene transcription, but in other areas of the genome, DNA methylation can help in recruiting proteins, which could lead to enhanced gene activity.

Histone modifications (acetylation, methylation, ubiquitination, and phosphorylation) can influence the way DNA is ordered and packaged and activate or repress gene transcription, as well as increase or decrease DNA accessibility.

Synthesis of non-coding RNAs (e.g., microRNAs, long non-coding RNAs) play an important role in the regulation of gene transcription and post-translational processing, acting as essential epigenetic regulators in a range of cellular and biological processes.

Epigenetic measures could be genome-wide or use a targeted approach focusing on specific genes. These need to be measured at the time of (for retrospective reporting of exposure) or after exposure, not prior to it. Additionally, to be eligible, studies needed to measure epigenetics in the offspring (first generation) or grandchildren (second generation) of individuals who were exposed to the trauma.

Data extraction: After removing duplicate articles, the title and abstract of all retrieved articles were first screened and, if eligible, the full text of the article was retrieved. For articles eligible for inclusion in the review, data were extracted using a standard data extraction form. The data extracted included the first author and year of publication, study design, characteristics of the participants who were exposed and those who measured the outcome, the type and details of the exposure, including assessment and timing, details related to the outcome measure, and summary findings.

## 3. Results

### 3.1. Characteristics of Included Studies

The database search yielded 314 non-duplicate articles in total. The process of selection of studies and the number included are shown in Figure 1 (adapted from the PRISMA flow diagram used in systematic reviews) [85].

We identified 22 publications from this scoping review, the first of which was published in 2011 [86]. Except the earliest two studies, all used the term intergenerational rather than transgenerational. Summary data from these publications are shown in Table 1. These publications come from 19 independent studies, as in some cases the same study has been used to investigate different genes [87,88,89,90,91].

Eighty-two percent of studies were undertaken in North America or Europe, and the sample size ranged from 21 to 896, with only 2 of these studies involving more than 201 participants [92,93]. The most commonly investigated exposure was childhood maltreatment in the parent, which was investigated, retrospectively, in nine studies. There were six studies of trauma or PTSD following war, including two studies of Holocaust survivors [83,94,95], two studies on the Rwandan genocide [90,91], and one each from the Kosovo War [96] and Vietnam veterans [97]. Two studies investigated interpersonal violence [86,98] and two investigated major depression [99,100]. There were also individual studies of other exposures (traumatic life events [101], psychological distress [102]). Most of the studies (82%) assessed past stress exposure, and it was thus collected retrospectively by self-report. In the majority of studies (n = 14), the exposures were assessed prior to conception. Four studies investigated pregnancy exposure [90,91,96,99], three investigated both pregnancy and pre-pregnancy exposure [86,100,103], and one study investigated parental exposure after pregnancy [102]. The majority of the studies looked at only maternal transmission, but four considered both maternal and paternal [83,94,95,102], and two focused only on fathers [97,104].

In terms of the epigenetic outcomes, all studies measured DNA methylation, which was assessed predominantly in the offspring’s blood (whole blood, umbilical cord blood), anywhere from birth to when the offspring were aged 19 years. However, some studies measured DNA methylation in buccal cells [87,99,102,105,106], saliva [98,103], or placenta [100]. One study measured DNA methylation in the father’s sperm [97]. There were six studies that measured genome-wide DNA methylation, one investigated epigenetic age [92], and the remainder used a candidate gene approach, and *NR3C1* (n = 6 studies) and *FKBP5* (n = 5 studies) were the most commonly studied genes.
genes-14-01639-t001_Table 1Table 1Studies investigating the association between trauma or stress and epigenetics in the next generation using the terms transgenerational or intergenerational.1st Author, YearStudy and Participant CharacteristicsExposure in G0 (Timing)Tissue, Epigenetic Measure, TimingFindings for Trauma-Exposed GroupTerm UsedRadtke, 2011 [86]25 mothers and children aged 10–19 years. GermanyIntimate partner violence, pre and during pregnancy (retrospective)Blood, 10–19 years. *NR3C1* DNA methylationIn pregnancy: higher methylation; pre-pregnancy: NSTransPerroud, 2014 [91]25 mothers of Tutsi ethnicity and 25 children aged 17–18 years. Rwanda (cases) or abroad (controls)Rwandan Genocide trauma when pregnant (retrospective)Blood, 17–18 years. *NR3C1* and *NR3C2* DNA methylationHigher methylationTransYehuda, 2014 [95]80 adult offspring (conceived after trauma) and 15 matched controls. USAHolocaust, PTSD (DSM-IV), maternal, and/or paternal (prospective)Blood, mean 57 years. *NR3C1* DNA methylationPaternal only: higher methylation; both parents: lower methylationInterStroud, 2016 [100]153 mothers and infants from a low-income diverse sample. USAPreconception and prenatal MDD (retrospective)Placenta *HSD11B2* DNA methylationNo significant overall associationsInterYehuda, 2016 [83]22 adult offspring and 9 matched controls. USAHolocaust, PTSD (DSM-IV), CTQ (prospective)Blood, mean 77 years. *FKBP5* DNA methylationHolocaust: lower methylation; CTQ: NSInterCimino et al., 2018 [102]21 families with children aged 6–10 years. ItalyPsychological distress mothers and fathers (prospective) Buccal swabs, 6–10 years.*DAT* DNA methylationHigher methylation InterMehta, 2019 [97]38 male Vietnam veterans, mean age 67, 16 with PTSD. Australia or NZWar-related PTSD, DSM-V (prospective)Sperm, genome-wide DNA methylation 3 CpG sites identifiedInterRamo-Fernandez, 2019 [88]113 mothers with newborn children. Germany Childhood maltreatment (retrospective)Umbilical cord blood fetal immune cells. *FKBP5, NR3C1, CRHR1* DNA methylationNo significant associationsInterBierer, 2020 [94]147 adult offspring (conceived after the trauma) and 31 controls. Mean age 51 years. USAHolocaust, in childhood or adulthood (retrospective)Blood, mean 51 years. *FKBP5* DNA methylation Exposure in childhood: Lower methylation; adulthood: NS InterGrasso, 2020 [103]114 women in 3rd pregnancy trimester and newborns (24 h after delivery), USAACEs and adult adversity (retrospective), prenatal PTSD symptoms (prospective)Newborn saliva, 24 h post-delivery. *FKBP5* DNA methylationHigher methylation with PTSD and threat-based ACEsInterPilkay, 2020 [101]201 mothers and newborns. USA20 traumatic life events (retrospective)Umbilical cord blood. *BDNF* DNA methylationFear and males only: higher methylation. InterHjort, 2021 [96]117 women and children aged mean 12 years. KosovoPTSD from sexual violence during war, pregnancy (retrospective)Blood, 6–18 years. Genome-wide DNA methylation Nominal CpGs but not when adjustedInterMerrill, 2021 [104]45 fathers and children aged 3 months. CanadaACEs (retrospective)Blood, 3 months. Genome-wide DNA methylation8 CpG sitesInterNwanaji-Enwerem, 2021 [92]238 women and children. Mexican–AmericanACEs up to 18 years (retrospective)Blood, 7, 9, and 14 years. Epigenetic age acceleration (8 measures)Age acceleration (Horvath and Intrinsic)InterRamo-Fernandez, 2021 [89]113 mothers and newborn children. Germany Childhood maltreatment (retrospective)Umbilical cord blood fetal immune cells. *OXTR* DNA methylationNo significant associationsInterCordero, 2022 [98]48 mothers and children aged 12–42 months. Switzerland.Interpersonal violence-related PTSD (retrospective)Saliva, 1–2 years.*NR3C1* DNA methylationHigher methylationInterFolger, 2022 [105]53 mother–child pairs, pregnancy and infant development study. USAACEs (retrospective)Buccal swabs, 1 month.Secretogranin V (*SCG5*) geneLower methylationInterMavioglu, 2022 [87]113 mother–newborn dyads, 1st week after birth. GermanyChildhood maltreatment (retrospective)Umbilical cord blood and buccal swabs (n = 68). *DNMT1* methylationNo significant associationsInterMoore, 2022 [106]Pregnant women <22 weeks gestation and 12-week infants (124 buccal, 92 blood). CanadaACEs (retrospective)Blood or buccal, 3 months. Genome-wide DNA methylation Numerous nominally but not adjusted significant CpGsInterMusanabaganwa, 2022 [90]33 mothers of Tutsi ethnicity and 26 children aged 17–18 years. Rwanda (cases) or abroad (controls) Rwandan Genocide trauma, pregnancy (retrospective)Blood, 17–18 years. Genome-wide DNA methylation16 differentially methylated regions identifiedInterMendonca, 2023 [99]60 mothers with children aged 6–12 years. BrazilMajor depression, pregnancy (retrospective)Buccal, 6–12 years. *FKBP5* and *NR3C1* DNA methylationLower FKBP5 methylationInterScorza, 2023 [93]896 mothers and infants just after birth. UKACEs, cumulative score (retrospective)Umbilical cord blood at birth. Genome-wide DNA methylation5 CpG sites, but only in male offspringInterPTSD: post-traumatic stress disorder. ACEs: adverse childhood experiences. CTQ: Childhood Trauma Questionnaire; DNMT1: DNA methyltransferase 1; NS: not significant.


### 3.2. Summary of Findings

Overall, 19 of 22 studies reported differential DNA methylation in offspring whose parents were exposed to stress/trauma. Of the studies investigating specifically *NR3C1*, four out of six reported higher DNA methylation in offspring after trauma exposure [86,91,95,98], one only reported pregnancy rather than prenatal exposure [86], and another reported mixed findings, depending on the parent who was exposed [95]. Of the five studies that measured *FKBP5* DNA methylation, three found lower methylation in the offspring of exposed parents [83,94,99], one found an opposite association with higher methylation [103], and another reported no association [88].

## 4. Discussion

### 4.1. Summary of Findings

Overall, the findings of the scoping review provide preliminary supporting evidence that parental exposure to trauma and stress may be passed to the next generation and alter DNA methylation patterns in the offspring.

A number of studies investigated maternal exposures in pregnancy, and these findings support the extensive prior work (including those summarized in Section 1.3.3), which highlights that the in utero environment can directly impact DNA methylation of the infant. However, 17 studies included here looked at exposures prior to conception and still provide some preliminary support for intergenerational transmission. While only two of these studies specifically looked at paternal transmission, both of these were epigenome-wide studies that identified significant sites that were differentially methylated in the blood [104] and sperm [97]. These latter findings are supported by work in animals, which showed the strongest evidence for male-line inheritance of stress. The findings are also supported by a small study that found that childhood abuse (physical, emotional, and/or sexual) was associated with differential DNA methylation at 12 regions in the sperm [107].

### 4.2. Limitations and Challenges

Limitations of the evidence presented here include that this was a scoping review to assess the extent to which studies have reported intergenerational or transgenerational inheritance of stress, with epigenetic mechanisms examined. We, therefore, have not included studies that are intergenerational in nature but have not used these explicit terms. As such, our scoping review provides a good representation of how these terms are being used in the field, but it is not exhaustive of all human studies in this area. For example, a study of grandmaternal psychosocial stress (interpersonal violence) during pregnancy found differential DNA methylation at five CpG sites in the offspring [108]. These sites were in genes previously involved in circulatory system processes.

The findings of the scoping review are also limited by the quality of the studies included. Many of these studies in humans include very severe trauma exposures in specific circumstances, and, therefore, it is unclear to what extent the findings can be applied more broadly. Furthermore, the findings are based on observational studies, and the vast majority of stress/trauma exposures were based on retrospective data. Recall bias resulting from differential reporting and recall of the exact type of exposure, as well as its severity and impact, will influence the findings. There are a wide range of other potential biases and limitations. Unlike animal studies, it is not possible to control for all external environmental factors or manipulate these with ease. It is thus impossible to fully tease apart the true causal effects of exposure. Indeed, despite much excitement in this area, caution must be taken to not over-interpret the current findings. The inability to detangle intergenerational transmission of environmental exposure via epigenetic modifications from other sources of parent–offspring transmission (including shared genetics, environment, and behaviors) will continue to be problematic and is something that cannot easily be resolved [109]. For example, the shared genetics between parents and offspring, as some individuals have been shown to be genetically more susceptible to the effects of stress and are also being exposed to trauma in the first place. Factors in the pre- and postnatal environment are also more similar between parents and offspring than across different families, and known as well as unknown factors may contribute to the likelihood of exposure to stress and trauma. Furthermore, parents who carry a legacy of trauma may differ in their behavior, and this could, in turn, influence the offspring. This highlights the possibility that the effects of trauma being passed to the next generation could occur via non-epigenetic mechanisms. Ultimately, only many more studies and the independent replication of key findings are needed to provide more solid evidence that stress and trauma can be transmitted across generations in humans via epigenetic effects.

### 4.3. Future Research Needed

Most studies have focused on candidate genes involved in HPA axis stress signaling, notably *NR3C1* and *FKBP5*. Only a few epigenome-wide association studies (EWAS) have been undertaken, where thousands of genetic loci are investigated simultaneously. The advantage of EWAS is that they allow for a thorough investigation of DNA methylation patterns across the genome, rather than targeting only genes thought to be implicated in trauma and stress. They, therefore, offer the distinct advantage of identifying new biological pathways implicated in pathophysiology and are thus targets for early intervention and treatment. The disadvantage of EWAS is that multiple testing increases the risk of false positive findings, and this is normally offset by applying stricter criteria for statistical significance. As a result, EWAS require very large sample sizes to detect small effects, and most current studies are likely to be underpowered.

The vast majority of research to date has focused on DNA methylation. Animal studies have shown histone modifications in association with anxiety, fear, or stress-related phenotypes, but they have received less attention in human studies of PTSD [31]. The role of non-coding RNA in PTSD is an emerging area of strong interest, with preliminary findings suggesting the role of non-coding RNAs in glucocorticoid signaling and dysfunction of the HPA axis and thus a critical role in PTSD pathophysiology [110]. Reduced levels of specific miRNAs have been found in a small sample of men exposed to early life stress [111]. Small non-coding RNAs (sncRNAs) have also been suggested to play an important role in transgenerational epigenetic inheritance [112]. Future research in this area needs to consider epigenetic marks beyond DNA methylation.

The exact role of epigenetics and the mechanisms leading to the transmission of trauma and stress across generations in humans remain unclear. Specific inheritance can occur through the germline (sperm and egg) cells, and the changes could remain through embryogenesis and fetal development [113]. Indeed, the epigenetic profile of gamete DNA can be altered by parental exposures, and the sperm epigenome can influence gene expression and development in the embryo [114]. During early embryo development, many epigenetic marks are erased, but there is now evidence that some of these may be resistant to reprogramming after fertilization [115]. They can thus be retained and inherited by the developing embryo. While there is quite good evidence from animal studies about the transmission of stress through the paternal line, epigenetic inheritance through the maternal line is also possible, but there has been limited research to date. Collecting germline cells has a number of obvious complications, particularly for the maternal line. Further research is also needed to determine the role of non-epigenetic gametic alterations.

More advanced studies are needed in this area that follow guidelines for improving study design and helping to ensure that high-quality findings are produced and that they are are robust and reproducible [109]. Studying the mechanisms of epigenetic inheritance in humans is challenging because of their observation nature. Due to likely contributions from a range of external factors, variations in findings may be expected, depending on the specific type and timing of trauma and stress. It is also difficult to track multigeneration cohorts that require very-long-term follow-up in humans.

## 5. Conclusions

There is considerable interest in the potential non-genetic transmission of a suite of stress-related health conditions, such that exposures occurring long before conception may have the capacity to impact subsequent generations. Epigenetics is emerging as a candidate mediator of such effects, in particular through epigenetic changes within germ cells and most commonly in the paternal line. Findings from animal studies are relatively strong in this regard, with only emerging and limited evidence to date in human studies. There are several challenges with such studies in humans and novel methods, and multifaceted integrative analyses are likely needed to further our understanding of the extent to which trauma and stress can become biologically inherited. There is an exciting potential that epigenetics may help explain some of the links between parental exposures and offspring health, thereby informing novel interventions, but future studies must address current limitations to advance translational knowledge in this area.

## Figures and Tables

**Figure 1 genes-14-01639-f001:**
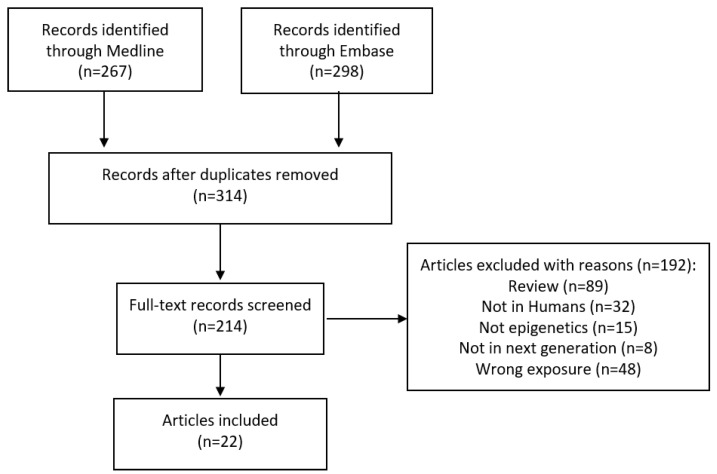
Limited PRISMA flow diagram.

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
