# Peer review of "Biological Embedding of Early-Life Adversity and a Scoping Review of the Evidence for Intergenerational Epigenetic Transmission of Stress and Trauma in Humans"

_genes, 2023, doi:10.3390/genes14081639_

Round 1

Reviewer 1 Report

The authors discussed the biology of early-life adversity and a scoping review of the evidence of intergenerational epigenetic transmission of stress and trauma in humans, which is officially called epigenetics and different from those epigenomic changes. This is an interesting review of the important environmental factors affecting human health and diseases. Still, it is far more complex due to the heterogeneity in both types of stress and individual coping strategies to mitigate the impact that the stress may induce. The manuscript is written well. I don’t have many critics of the manuscript, but the problems might be the original studies.

 1.       There may be helpful if a brief definition or classification of stress is provided. We know stressful events can be classified into acute vs. chronic, perceived vs. unpredictable, and traumatic. These may have different impacts on mothers and infants. For example, social stress and chronic stress due to financial difficulty or poverty in low-income countries may cause mother’s or malnutrition in mothers and then increase the risk of maternal infections, which might be more prevalent and have a greater impact on infants than those traumatic stress. It is well-known that poverty is linked to infectious diseases such as tuberculosis.

2.       Most of these studies were based on retrospective data; they must have recall bias, and it seemed the traumatic events were inconsistently measured, and therefore, results seemed quite different.

3.       Line 79-80. “….More recently, ….”. Barkers’s hypothesis has been more than three decades and DoHad has been nearly twenty.

Author Response

The authors discussed the biology of early-life adversity and a scoping review of the evidence of intergenerational epigenetic transmission of stress and trauma in humans, which is officially called epigenetics and different from those epigenomic changes. This is an interesting review of the important environmental factors affecting human health and diseases. Still, it is far more complex due to the heterogeneity in both types of stress and individual coping strategies to mitigate the impact that the stress may induce. The manuscript is written well. I don’t have many critics of the manuscript, but the problems might be the original studies.

  1. There may be helpful if a brief definition or classification of stress is provided. We know stressful events can be classified into acute vs. chronic, perceived vs. unpredictable, and traumatic. These may have different impacts on mothers and infants. For example, social stress and chronic stress due to financial difficulty or poverty in low-income countries may cause mother’s or malnutrition in mothers and then increase the risk of maternal infections, which might be more prevalent and have a greater impact on infants than those traumatic stress. It is well-known that poverty is linked to infectious diseases such as tuberculosis.

Response 1: This is a very valid point. We have now clearly defined psychological stress at the start of the review (lines 31-35). We have also included additional text at the end of the introduction, where we mention that the impact of stress could be through pathways and mechanisms unrelated to the stress response system, as the reviewer mentions (additional text added lines 70-76).

  1. Most of these studies were based on retrospective data; they must have recall bias, and it seemed the traumatic events were inconsistently measured, and therefore, results seemed quite different.

Response 2: The reviewer is correct on both accounts. In regards to the retrospective reporting of the stress/trauma, we have noted this in Table 1. However, in the results we have added specific mention of the percentage of studies with retrospective data (page 9, line 432), and have added this as a limitation in the Discussion (page 12, lines 502-504).

  1. Line 79-80. “….More recently, ….”. Barkers’s hypothesis has been more than three decades and DoHad has been nearly twenty.

Response 3: We have corrected this and removed the “More recently” at line 78.

Reviewer 2 Report

In this review, the authors undertook a scoping review to determine the extent to which the terms intergenerational or transgenerational have been used in human studies investigating the transmission of trauma and stress via epigenetic mechanisms. This review article is well written and contains up-to-date literature survey and has significant impact.

Only a few minor concerns:

Title name of 1.1 “ stress increases...” seems inadequate.

In utero, line 114 and others, should be in italic font;

NR3C1 or Nr3c1; BDNF or Bdnf; FKBP5 or Fkbp5, please be consistent,

Line 102, CVD stands for what? Line 151, Bdnf should have full a name here; Table 1, what are CTQ and DMR stand for?

A few typos were found: line 156, wrong acronym for CRH receptor, not Crf/Crh; line 307, [81] [81] ...[77] [77]; line 530, HPA only; line 619, be BMJ; line 805, be 1046;

Author Response

In this review, the authors undertook a scoping review to determine the extent to which the terms intergenerational or transgenerational have been used in human studies investigating the transmission of trauma and stress via epigenetic mechanisms. This review article is well written and contains up-to-date literature survey and has significant impact.

Only a few minor concerns:

Title name of 1.1 “ stress increases...” seems inadequate.

Response 1: We have changed the subheading at line 31 to “1.1 Stress as a major risk factor for a range of non-communicable diseases.”.

In utero, line 114 and others, should be in italic font;

Response 2: This correction on current lines 85, 121, 228, and 306 have been made.

NR3C1 or Nr3c1; BDNF or Bdnf; FKBP5 or Fkbp5, please be consistent,

Response 3: We have followed proper gene nomenclature which stipulates that human genes needs to be capitalized, whereas those for mouse and rat have only the first letter in uppercase and the remaining letters in lowercase. Hence why we need to have both NR3C1 and Nr3c1 (the latter specifically in Sections 1.3.1 and 1.4.2 when referring to the animal studies) in the article.

Line 102, CVD stands for what? Line 151, Bdnf should have full a name here; Table 1, what are CTQ and DMR stand for?

Response 4: We have spelled out CVD and Bdnf in full. CTQ is now defined in the footnote, and DMR is spelled out in full in the Table.

A few typos were found: line 156, wrong acronym for CRH receptor, not Crf/Crh; line 307, [81] [81] ...[77] [77]; line 530, HPA only; line 619, be BMJ; line 805, be 1046;

Response 5: We have made corrections to these typos. As mentioned in response 3 above, in rats the gene name is Crh (it can also be called Crf, but we have removed that to avoid confusion). We have removed the repeated reference on current line 314. We have removed the full definition of HPA on current line 543. We have corrected the 2 typos in the reference list.